# AutoCleansing: Unbiased Estimation of Deep Learning with Mislabeled Data

## Abstract

Mislabeled samples cause prediction errors. This study proposes a solution to the problem of incorrect labels, known as **AutoCleansing**, to automatically capture the effect of incorrect labels and mitigate it without removing the mislabeled samples. AutoCleansing consists of a base network model and sample-category specific constants. Both parameters of the base model and sample-category constants are estimated simultaneously using the training data. Thereafter, predictions for test data are made using a base model without the constants capturing the mislabeled effects. A theoretical model for AutoCleansing is developed and showing that the gradient of the loss function of the proposed method can be zero at true parameters with mislabeled data if the model is correctly constructed. Experimental results show that AutoCleansing has better performance in test accuracy than previous studies for CIFAR-10, CIFAR-100, SVHN, and ImageNet datasets.

## 1 Introduction

The prediction performance of supervised machine learning depends on the quality of the training data. For classification tasks, the dataset is assumed to have a correct label for each object. However, real-world datasets may contain some mislabeled samples. For instance, Pleiss et al. (2020) analyzed incorrect labels in the CIFAR-10 and CIFAR-100 datasets (Krizhevsky & Hinton, 2009). They reported that the mislabeled sample was 3 % of CIFAR-10 and 13 % of the CIFAR-100 datasets.

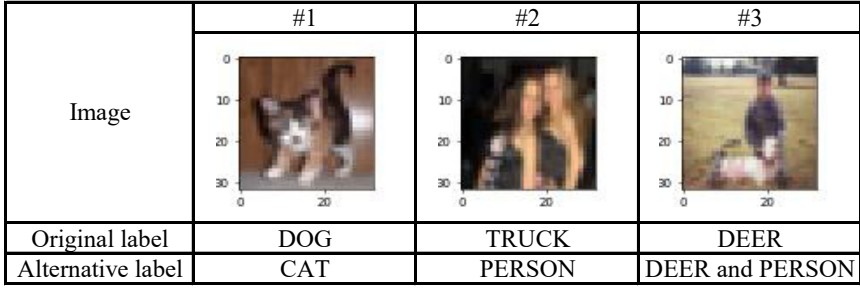

| | #1 | #2 | #3 |
|---|---|---|---|
| Image | | | |
| Original label | DOG | TRUCK | DEER |
| Alternative label | CAT | PERSON | DEER and PERSON |

Figure 1: Example of incorrect labels in CIFAR-10. The original label is the corresponding label of each image in the dataset. Alternative label is the possibly correct label of each image.

Figure 1 shows typical examples of incorrect labels in the CIFAR-10 dataset. It consists of 60,000 images in 10 category classes. Each image was assigned one of 10 classes. In this figure, the original label of #1 is DOG; however, it appears to be the image of CAT. As the category set of CIFAR-10 includes both DOG and CAT, #1 is considered to be an example of **an incorrect label within the category set**. The image of #2 has TRUCK as the original label. However, it shows the image of PERSON, which does not belong to the category set. Thus, this is an example of **an incorrect label outside the category set**. For the image of #3, there are two objects in this image; however, it has only one label of DEER. It can be considered as an example of **an incorrect label with multiple objects**.

Incorrect labels in the training dataset may cause prediction errors. The most intuitive way to address the problem of incorrect labels is by removing mislabeled samples from the training dataset. However, to identify mislabeled samples, it is necessary to measure the correctness of labels and define the threshold determining whether the label is correct or not. Deleting excess data may reduce the efficiency of estimation by decreasing the sample size. Finding an optimal threshold requires several runs of learning by removing mislabeled samples with different thresholds.

This study proposes an alternative solution to the problem of incorrect labels, called **AutoCleansing**, to automatically capture the effect of incorrect labels and mitigate it without removing mislabeled samples. AutoCleansing consists of a base network model and sample-category specific constants. Both parameters of the base model and sample-category constants are estimated simultaneously using the training data. Thereafter, predictions for test data are made using a base model without the constants capturing the mislabeled effects.

Figure 2 shows the concept of AutoCleansing. Let $x$ be the input, $y$ be the output, and $y = m(x, \theta)$ be the base network model, where $\theta$ denotes the parameter of the base model. Consider five observations of $A$, $B$, $\cdots$, $E$. The red line is the true model defined as, $y = m(x, \theta^*)$ where $\theta^*$ denotes the true parameter. $B$ is a mislabeled sample, as the observed label $B$ differs significantly from the true label $B^*$. The dotted line is the estimated model $y = m(x, \hat{\theta})$ using incorrect data, where $\hat{\theta}$ denotes the estimated parameter. As can be observed, overfitting occurs owing to the mislabeled sample. In this figure, $\hat{y}$ denotes the prediction for $x = 3$ using the estimated model; however, the true label is $y^*$. Thus, the incorrect label causes the prediction error. Consider the cleansing model, $y = m(x, \theta) + \alpha$, where $\alpha$ denotes the constant parameter for each observation. If the constant $\alpha_B$ captures the effect of an incorrect label, as shown in this figure, removing the constant from the cleansing model may mitigate the overfitting problem.

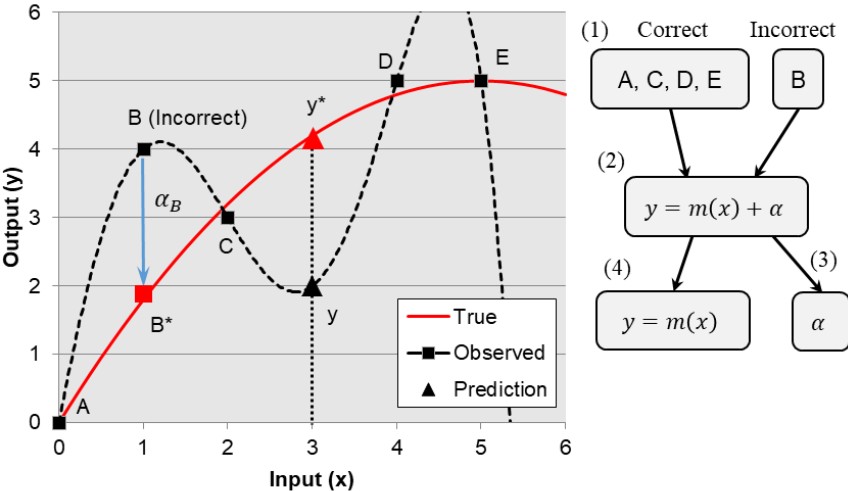

Figure 2: Concept of AutoCleansing. (Left) $A$ to $E$ are the observations. $B$ is the incorrect label. Red line represents the true model. Black dot line represents the estimated model using incorrect data. $y$ denotes the predicted label using the overfitting model. $y^*$ denotes the true label. AutoCleansing consists of the base network model and constant $\alpha$. The constant $\alpha_B$ captures the effect of the incorrect label for $B$. Thus, removing $\alpha$ mitigates the overfitting effect due to the incorrect label. (Right) (1) Training data has correct and incorrect labels. (2) Constructing the cleansing model consists of a base network model $m(x)$ and sample-category specific constant $\alpha$. Learning with the training data using cleansing, (3) Deleting the constant $\hat{\alpha}$, and (4) Testing with the validation data using the cleansed network model.

As shown in the section of the theoretical analysis, the proposed AutoCleansing can address the prediction errors due to the incorrect labels within the category set, outside the category set, and with multiple objects. AutoCleansing can use any network model as the base model with any augmentation method. For example, the experimental section presents the estimation results of AutoCleansing with base models of ResNet (He et al., 2016), WideResNet (Zagoruyko & Komodaki, 2016), Shake-

Shake (Gastaldi, 2017), and PyramidNet+ShakeDrop (Yamada et al., 2018) using AutoAugment (Cubuk et al., 2018). Furthermore, AutoCleansing does not require iterative runs to identify the incorrect labels because the effects of the incorrect labels are automatically captured in a single run by the sample-category specific constant.

The contribution of this study can be summarized as follows:

- It provides a theoretical model for AutoCleansing. The incorrect labels in the training data cause a prediction error. The proposed method can capture the biased effects of incorrect labels automatically and address the problem of prediction error due to incorrect labels.
- The proposed method can be implemented with any network model or augmentation method. This study considers experiments of AutoCleansing with ResNet, WideResNet, Shake-Shake, and PyramidNet+ShakeDrop using AutoAugment.
- Experimental results show that the proposed AutoCleansing method can improve the validation accuracy for the CIFAR-10, CIFAR-100, SVHN, and ImageNet datasets.
- The additional training cost of AutoCleansing relative to the base network model can be negligible. AutoCleansing can remove the biased effect of incorrect labels automatically in a single run of learning. For example, the additional learning time of AutoCleansing with CIFAR-10/100 datasets is only 0.5 % of that of the base network models.

## 2    RELATED WORKS

There are several studies on noisy datasets in the literature on machine learning. Frnay & Verleysen (2014) provide a comprehensive review of label noise in classification. Algan & Ulusoy (2019) provide a complete overview of deep learning with noisy datasets. There are three approaches to dealing with mislabeled datasets as follows: (1) robust learning with label noise, (2) identification of mislabeled data, and (3) utilization of a small dataset without incorrect labels.

For robust learning with label noise, after the early works of Reed et al. (2015) and Azadi et al. (2015), several algorithms using deep neural networks have been proposed including the S-model (Goldberger & Ben-Reuven, 2016), MentorNet (Jiang et al., 2018), decoupling (Malach & Shalev-Shwartz, 2017), F-correction (Patrini et al., 2017), Open-set (Wang et al., 2018), Bi-level-model (Jenni & Favaro, 2018), Lq (Zhang & Sabuncu, 2018), co-teaching (Han et al., 2018), random reweighting (Ren et al., 2018), joint optimization (Tanaka et al., 2018), DAC (Thulasidasan et al., 2019), SELF (Nguyen et al., 2020), dynamic bootstrapping (Arazo et al., 2019), and DivideMix (Li et al., 2020). Goldberger & Ben-Reuven (2016) and Patrini et al. (2017) estimated a noise transition matrix to correct for the loss function. However, it was difficult to correctly estimate the transition matrix. Jiang et al. (2018) and Ren et al. (2018) proposed weighted samples to adapt the noisy samples. However, estimation of correctly weighted samples was also challenging. Arazo et al. (2019) proposed a beta mixture model of the cross-entropy loss of each sample and modeled the label noise. Their approach showed outstanding performance for high-level noise. For linear regression, a consistent robust regression was proposed for the corrupted data (Bhatia et al., 2017). However, the consistency of robust learning method of nonlinear estimation using the classification model was not clear. These studies of robust learning with label noise used datasets having synthetic-label noise added. Label noise is generated by replacing one label with another at a given probability within a category set. These studies showed good performance for artificially generated label noise. However, most did not find mislabeled samples in real-world datasets. On the other hand, the approach proposed in the this study considers incorrect labels within and outside the category set as well as multiple objects in real-world datasets.

For the identification of mislabeled data, some studies found incorrect labels in famous datasets for deep learning. For instance, Ekambaram et al. (2017) found 92 mislabeled examples in 18 of the image classes, after reviewing more than 15 % of the images in the ImageNet dataset. Al-Rawi & Karatzasu (2018) reported 9 incorrect samples on CIFAR10 and 15 mislabeled samples on CIFAR100. Müller & Markert (2019) reported 4 mislabeled samples on MNIST, 7 samples on CIFAR100, and 64 samples on Fashion-MNIST. Pleiss et al. (2020) identified incorrect labels using the area under the margin (AUM) statistic. They showed that incorrect labels are 3 % on CIFAR10, 13 % on CIFAR100, and 24 % on Tiny ImageNet. On CIFAR100, they reported that removing 13 % of the data leads to a 1.2 % drop in error. Identifying mislabeled data requires some criteria

to determine whether it is correct or incorrect. Several runs of learning may be required to search for optimal criteria for the identification of incorrect labels. The method proposed in this study, in contrast, allows us to capture and remove the effects of incorrect labels in a single run of learning.

For the utilization of a small dataset without incorrect labels, it is assumed that we have a small set of clean data, namely, free of mislabeled samples. Sukhbaatar et al. (2015) and Hendrycks et al. (2018) used clean data to estimate the noise-transition matrix for incorrect labels. Other studies on the utilization of clean data include Ren et al. (2018), Li et al. (2017), and Zhang et al. (2019b). However, it is difficult to find a small set of clean data in real-world datasets. Meanwhile, the method proposed in this study does not require clean data.

## 3 AUTOCLEANSING

Consider the classification task with $K$ categories for the $N$ training data points $X = \{x_1, x_2, \cdots, x_N\}$ and labels $Y = \{y_1, y_2, \cdots, y_N\}$. Let the learning network model be, $M(x, \theta) = \{m_1(x, \theta), \cdots, m_K(x, \theta)\}$ that assigns an input $x$ to an output $m$ with a given parameter $\theta$. The predicted probability of output $y$, given the input $x$ using this model, is assumed to be calculated by the following softmax function:

$$P(y = i|x) = \frac{e^{m_i}}{\sum\limits_{j \in \mathbb{K}} e^{m_j}} \tag{1}$$

where $m_i = m_i(x, \theta)$ denotes the $i$th element of output and $\mathbb{K} = \{1, \cdots, K\}$ the category set.

If the training data has mislabeled samples, the estimated parameter might be biased, which could cause a prediction error. To address the problem of incorrect labels, consider the cleansing network model, $C(x, \theta) = M(x, \theta) + \alpha$, where $\alpha \in \mathbb{R}^{N \times K}$ is a parameter of the sample-category specific constant. Let $\alpha_k$ be the constant for input $x$ of the sample and category $k$. Therefore, the prediction probability $P^C$ with the constant can be expressed as follows:

$$P^C(y = i|x) = \frac{e^{m_i + \alpha_i}}{\sum\limits_{j \in \mathbb{K}} e^{m_j + \alpha_j}} \tag{2}$$

If the effect of the mislabeled samples is captured by the sample-category-specific constant $\alpha$ of the cleansing network model $C(x, \theta, \alpha) = M(x, \theta) + \alpha$, the base network model $M(x, \theta)$ could avoid the biased problem due to incorrect labels. The learning process with AutoCleansing is as follows: (1) Learning with the training data using a cleansing network model $C(x, \theta, \alpha) = M(x, \theta) + \alpha$, (2) deleting the sample-category-specific constant $\hat{\alpha}$ estimated in the learning process, and (3) testing with the validation data using a cleansed network model, $C(x, \hat{\theta}, \hat{\alpha}) - \hat{\alpha}$, where $\hat{\theta}$ denotes the estimated parameter of the base model.

### 3.1 THEORETICAL ANALYSIS OF AUTOCLEANSING

To investigate the performance of the proposed AutoCleansing for mislabeled data, we consider the following definition of a general case:

**Definition 1** (Incorrect labels and outside of the category set). *Let $\mathbb{K}^*$ be a full set of true categories and $\mathbb{K} \subset \mathbb{K}^*$ be the category set of the model. Let $\pi(\hat{y}|y^*x)$ be the probability such that the input $x$ with the true category $y^* \in \mathbb{K}^*$ has label $\hat{y} \in \mathbb{K}$. Let $P(y^* \mid x, \mathbb{K}^*, \theta^*)$ be the true probability of the category $y^*$ from the category set $\mathbb{K}^*$ with input $x$ given true parameter $\theta^*$. Thus, **the observed probability of category $\hat{y} \in \mathbb{K}$ for the input $x$ with incorrect labels within and outside of the category set** is defined as follows:*

$$Q\left(\hat{y} \mid x, \mathbb{K}^*, \theta^*\right) = \sum_{y^* \in \mathbb{K}^*} \left[\pi\left(\hat{y}|y^*, x\right) \cdot P\left(y^* \mid x, \mathbb{K}^*, \theta^*\right)\right]$$

This definition includes incorrect labels, outside of the category set, and multiple objects. If $\mathbb{K}^* = \mathbb{K}$, this definition is equivalent to that within the category set. If incorrect labels occur outside the category set, namely $\pi(\hat{y}|y^*, x) = 0 \, \forall \hat{y} \neq y^* \in \mathbb{K}$ and $\pi(\hat{y}|y^*, x) \geq 0 \, \forall y^* \in \mathbb{K}^* \setminus \mathbb{K}$, it defines the observed probability outside the category set. Let $S$ be the combination of categories in multiple

objects (for example, DEER and PERSON) and $\mathbb{K}^+ = \mathbb{K} \bigcup S$ be the true category set including the original categories and combination of categories in multiple objects. If $\mathbb{K}^* = \mathbb{K}^+$, definition 1 describes multiple objects.

If the sample data has incorrect labels within the category set, incorrect labels outside of the category set, or multiple objects, the estimated parameter using the minimum loss function does not converge to the true parameter; thus, the incorrect labels cause the prediction error. However, the following theorem shows that AutoCleansing can address the biased estimation due to mislabeled data:

**Theorem 1** (AutoCleansing for incorrect labels within and outside *the category set*). *Let $\mathbb{K}^*$ be a full set of true categories and $\mathbb{K}$ be the category set of the model. Let $\pi(\hat{y}|y^*, x)$ be a probability such that the input $x$ with the true category $y^* \in \mathbb{K}^*$ has label $\hat{y} \in \mathbb{K}$. Assume that the sample has incorrect labels and the outside of the category set is defined as Definition 1. Let $L^C$ be the expected loss function of AutoCleansing and $\theta^+$ be the set of the solution to $\partial L^C / \partial \theta = 0$. Furthermore assume that the model is correctly constructed and the probability distribution of the output is the softmax function. Then the gradient of the expected loss function with AutoCleansing is zero at the true parameter value $\theta^*$. Namely, $\theta^* \in \theta^+$,*

The proof can be found in the Appendix A.1. Note that, although this theorem states that the stochastic gradient descent using the loss function of the correct model with AutoCleansing can be stopped at true values, it does not guarantee that minimization of the loss function will converge to the true value, if the loss function has more than one local minimum. Theorem 1 shows that the implementation of the sample-category specific constant $\alpha$ can capture the biased effect of incorrect labels. This suggests that the value of $\alpha$ may reflect the effects of incorrect labels. The following theorem confirms this:

**Theorem 2** (Sample-category specific constants and incorrect labels). *Assume that the sample has incorrect labels and outside of the category set defined as Definition 1. Consider that the true label $t$ is assigned a false label $f$. Let $\hat{\alpha}_t$ and $\hat{\alpha}_f$ be parameters of the sample-category-specific constants of the true and false labels, respectively, which are estimated by the minimum loss function with AutoCleansing. Assume that the probability of observing a false label is greater than or equal to that of the true label, namely $Q(f \mid x, \mathbb{K}^*, \theta^*) \geq Q(t \mid x, \mathbb{K}^*, \theta^*)$. Furthermore assume that the model is correctly constructed, the probability distribution of the output is the softmax function, and the loss function is minimized at true parameter values. Then, as $N \to \infty$, the estimated parameter of the sample-category specific constants $\hat{\alpha}$ using the minimum loss function with AutoCleansing has the following properties* :

1. ***General case:** The sample-category-specific constants of the true label are equal to or less than that of the false label: $\hat{\alpha}_t \leq \hat{\alpha}_f$.*

2. ***Symmetric case:** Assume $\pi(f|t, x)$ is symmetric and independent of $x$, namely, $\pi(f|t, x) = \pi(f|t) = \pi(t|f)$. For this case, the sample-category specific constants of the true label are equal to or less than zero: $\hat{\alpha}_t \leq 0$.*

3. ***Single symmetric case:** Assume $\pi(f|t, x)$ is symmetric, independent of $x$, and incorrect labels occur between $t$ and $f$ only. Consequently, $\pi(f|t, x) = \pi(f|t) = \pi(t|f)$ and $\pi(j|j, x) = 1, \forall j \neq t, f$. For this case, the sample-category-specific constants of labels except the true and false labels are equal to zero: $\hat{\alpha}_j = 0, \forall j \neq t, f$.*

The Appendix A.2 provides the proof of this theorem. Table 1 provides numerical examples of the theorem. Assume that the category set has three categories $\mathbb{K} = \{1, 2, 3\}$. Example (A) is the incorrect label within the category set. Assume the true model output is $\{m_1^*, m_2^*, m_3^*\} = \{0.1, 0.1, 0.8\}$. Namely, the correct category is Category 3, having the highest output value. If the first label is observed, the incorrect label causes biased learning: $\{c_1, c_2, c_3\} = \{0.8, 0.1, 0.1\}$. The estimated sample-category-specific constants are assumed to be optimal such that $\alpha = m^* - c$. The constant of the true label ($\alpha_3 = -0.7$) is less than that of the false label ($\alpha_1 = 0.7$), such that the output value of the biased model ($c = m^* + \alpha$) of the observed category (0.8) is larger than the one of the true category (0.1). Example (B) is the incorrect label outside the category set. Consider that the true category set is $\mathbb{K}^* = \{1, 2, 3, 4\}$. Assume the true category is Category 4 outside of the observed category set. The optimal constant of observed label ($\alpha_1 = 0.7$) has the largest value. Note that the constant of true category ($\alpha_4 = -0.7$) is not estimated, because this category is the outside of category set. Example (C) is the multiple objects. Assume the observed label is 1, whereas the

true categories are 1 and 3. The constant of unobserved true category ($\alpha_3 = -0.7$) is less than that of the observed true label ($\alpha_1 = 0.0$),

Table 1: Neumerical example of AutoCleansing. *Obs.* is the observed label and *True* is the true category. *Outside* is the outside of the category set. $c$ is the output of biased model, $m^*$ is the output of true model, and $\alpha$ is the biased effects estimated by AutoCleansing.

| | (A) Incorrect label | | | (B) Outside of category set | | | | (C) Multiple objects | | |
|---|---|---|---|---|---|---|---|---|---|---|
| Label | 1 | 2 | 3 | 1 | 2 | 3 | 4 (outside) | 1 | 2 | 3 |
| | Obs. | | True | Obs. | | | True | Obs. and True | | True |
| $m^*$ | 0.1 | 0.1 | 0.8 | 0.1 | 0.1 | 0.1 | 0.7 | 0.8 | 0.1 | 0.8 |
| $c$ | 0.8 | 0.1 | 0.1 | 0.8 | 0.1 | 0.1 | 0.0 | 0.8 | 0.1 | 0.1 |
| $\alpha$ | 0.7 | 0.0 | -0.7 | 0.7 | 0.0 | 0.0 | -0.7 | 0.0 | 0.0 | -0.7 |

## 4 EXPERIMENTS

This section provides the experiments investigating the performance of the proposed AutoCleansing on the CIFAR-10 and CIFAR-100 (Krizhevsky & Hinton, 2009), SVHN (Netzer et al., 2011), and ImageNet (Russakovsky et al., 2015) datasets. The proposed AutoCleansing method requires base network models. In these experiments, the base network models are Wide-ResNet 40-2 and Wide-ResNet 28-10 (Zagoruyko & Komodaki, 2016), Shake-Shake 26 2x32d, 26 2x96d and 26 2x112d (Gastaldi, 2017), and PyramidNet+ShakeDrop (Yamada et al., 2018). All hyperparameters of base network models are the same as those used in AutoAugment (Cubuk et al., 2018), FastAutoAugment (Lim et al., 2019), and PBA (Ho et al., 2019)[1]. A cosine learning decay with one annealing cycle was applied to all models except ResNet. For AutoCleansing, the sample-category specific constants could not converge without regularization. The weight decay of the sample-category specific constant is $5 \times 10^{-5}$, except for PyramidNet, which uses $1 \times 10^{-5}$.

AutoCleansing has learning parameters of the sample-category specific constant $\alpha$ that consists *of* $K$ variables for each sample. Note that all $K$ variables cannot be identified; therefore, the constants for the first category are set to zero for all samples ($\alpha_1 = 0$). Thus, the estimated sample-category specific constant $\alpha \in \mathbb{R}^{N \times K}$ has $N(K-1)$ estimable parameters. In this study, the AutoCleansing with the sample-category-specific constant of $N(K-1)$ parameters is called as AC1. However, it might be difficult to estimate all parameters of $\alpha$ for large datasets. For example, ImageNet has more than 1.2 million images with 1,000 categories for training data. Therefore, AC1 needs to estimate more than 1.2 billion parameters. Instead, consider the sample specific constant $\alpha \in \mathbb{R}^N$ such that all categories except the observed label are set to zero for all samples ($\alpha_j = 0 \ \forall j \neq \hat{y}$); that is, $\alpha$ has $N$ estimable parameters. The AutoCleansing with the sample-specific constant of $N$ parameters is called as AC2. Note that AC2 corresponds to the special case of the single symmetric case of Theorem 2. For the single symmetric case, $\alpha_{nj} = 0 \ \forall n, \forall j \neq t, f$. If the true category belongs to the outside of the category set, we cannot estimate the $\alpha_{nt}$ of the true category, therefore, all categories except the observed label have zero values of $\alpha_{nj}$.

The experiments compare the results with baseline preprocessing, Cutout (DeVries & Taylor, 2017), AutoAugment (AA), FastAutoAugment (FAA), and Population Based Augmentation (PBA). The baseline preprocessing is conventional augmentation as follows: standardizing the data, horizontal flipping with 50 % probability, zero-padding, and randomly cropping. In the proposed AutoCleansing, we follow the procedure of AutoAugment, which first applies the baseline preprocessing, then applies the AutoAugment policy, and finally applies the Cutout.

### 4.1 EXPERIMENTAL RESULTS

The CIFAR-10 dataset has a total of 60,000 images, including 50,000 for training set and 10,000 for test sets. The number of categories was 10. Thus, the sample-category specific constant for AC1 has 0.45 million estimable parameters, whereas the sample specific constant for AC2 has 0.05 million

---

[1] See Table 7 in Appendix

estimable parameters. Table 2 shows the results of the test accuracy for different network models using the CIFAR-10 dataset. For all models, the proposed AutoCleansing with AutoAugment can achieve better performance compared to previous models[2].

Table 2: Test accuracy (%) on CIFAR-10. AC1+AA are the results of the proposed AutoCleansing with sample-category specific constants and AutoAugment. All experiments in this study replicate the results of Baseline, Cutout, and AutoAugment methods from Cubuk et al. (2018), FAA from Lim et al. (2019), and PBA from Ho et al. (2019). Averages of five runs are reported.

|  | Baseline | Cutout | AA | FAA | PBA | AC1+AA | |
| --- | --- | --- | --- | --- | --- | --- | --- |
| Wide-ResNet-40-2 | 94.70 | 95.90 | 96.30 | 96.30 | - | **96.56** | ±0.11 |
| Wide-ResNet-28-10 | 96.13 | 96.92 | 97.32 | 97.30 | 97.42 | **97.53** | ±0.08 |
| Shake-Shake (26 2x32d) | 96.45 | 96.98 | 97.53 | 97.50 | 97.46 | **97.60** | ±0.07 |
| Shake-Shake (26 2x96d) | 97.14 | 97.44 | 98.01 | 98.00 | 97.97 | **98.12** | ±0.11 |
| Shake-Shake (26 2x112d) | 97.18 | 97.43 | 98.11 | 98.10 | 97.97 | **98.27** | ±0.05 |
| PyramidNet+ShakeDrop | 97.33 | 97.69 | 98.52 | 98.30 | 98.54 | **98.59** | ±0.05 |

Table 3: Test accuracy (%) on CIFAR-100. Averages of five runs are reported.

|  | Baseline | Cutout | AA | FAA | PBA | AC1+AA | |
| --- | --- | --- | --- | --- | --- | --- | --- |
| Wide-ResNet-40-2 | 74.00 | 74.80 | 79.30 | 79.40 | - | **79.94** | ±0.20 |
| Wide-ResNet-28-10 | 81.20 | 81.59 | 82.91 | 82.70 | 83.27 | **84.07** | ±0.18 |
| Shake-Shake (26 2x96d) | 82.95 | 84.00 | 85.72 | 85.40 | 84.69 | **86.19** | ±0.12 |
| PyramidNet+ShakeDrop | 86.01 | 87.81 | 89.33 | 88.30 | 89.06 | **89.40** | ±0.11 |

Table 4: Test accuracy (%) on SVHN

|  | Baseline | Cutout | AA | FAA | PBA | AC1+AA |
| --- | --- | --- | --- | --- | --- | --- |
| Wide-ResNet-28-10 | 98.50 | 98.70 | 98.93 | 98.90 | 98.82 | **98.96** |

The CIFAR-100 dataset also has a total of 60,000 images, including 50,000 for training set and 10,000 for test sets. The number of categories was 100. Thus, the sample-category specific constant for AC1 has 4.95 million estimable parameters, whereas the sample specific constant for AC2 has 0.05 million estimable parameters. Table 3 provides the results for the CIFAR-100 dataset. Similarly, for CIFAR-10, the proposed model has better accuracy than previous models.

The SVHN dataset has 73,257 digit images for the core training set, 531,131 for the additional training set, and 26,032 for the test set. In this experiment, both core and additional training sets were used. The number of categories was 10. Thus, the sample-category specific constant for AC1 has 5.44 million estimable parameters, whereas the sample specific constant for AC2 has 0.53 million estimable parameters. Table 4 reports the results for the SVHN dataset. For the SVHN dataset, the proposed model has better accuracy than previous models.

The ImageNet dataset has more than 1.2 million images for the training set and 0.15 million images for the validation and test sets. The number of categories was 1,000. Thus, the sample-category specific constant for AC1 has more than 1.28 billion parameters that may not be feasible to estimate. Therefore, this experiment uses the sample specific constant for AC2, which has 1.28 million estimable parameters. Table 5 shows the results for the ImageNet dataset. The proposed model of AC2 has better Top1 accuracy than previous models, whereas the Top5 accuracy of AC2 is less than that of AA and FAA.

Pleiss et al. (2020) proposed the the area under the margin (AUM) statistic for robust learning with label noise. They provided experiments of label noise using real-world datasets and the artificially generated noise. Their experimental results showed that the AUM had better performance than

---

[2]Table 8 in Appendix provides a comparison between AC1 and AC2. Note that the test accuracy of AC1+AA is close to that of AC2+AA.

Table 5: Top 1 / Top 5 test accuracy (%) on ImageNet

|  | Baseline | AA | FAA | AC2+AA |
|---|---|---|---|---|
| ResNet-50 | 75.30 / 92.20 | 77.63 / **93.82** | 77.60 / 93.70 | **77.71** / 93.58 |

Table 6: AutoCleansing and Area Under the Margin (AUM). Base network model is ResNet32. This table replicates the results of Baseline and AUM from Pleiss et al. (2020). Averages of five runs are reported.

|  | Baseline | AUM | AC1 | |
|---|---|---|---|---|
| CIFAR-10 | 91.9 | 92.1 | **92.75** | ±0.06 |
| CIFAR-100 | 67.0 | 68.2 | **69.13** | ±0.24 |

those of the previous studies for datasets having synthetic label noise added. Table 6 shows the test accuracy of AutoCleansing and AUM using the ResNet 32 model. All hyperparameters of the base network models were the same as those used by Pleiss et al. (2020). AutoCleansing demonstrates outperforming the AUM on both CIFAR-10 and CIFAR-100.

## 4.2 Detection of Incorrect Labels using AutoCleansing

As shown in the theoretical analysis section, AutoCleansing can capture the effect of incorrect labels using the sample-category specific constant $\alpha$. If there is no mislabeled sample, $\alpha \to 0$, as $N \to \infty$. A large value of $|\alpha|$ indicates the existence of mislabeled samples in the data. Therefore, it might be possible to identify incorrect labels using the estimated sample-category specific constant $\hat{\alpha}$.

Figure 4-6 in Appendix show the sample-category specific constants $\hat{\alpha}$ for the example of mislabeled images in CIFAR-10 and CIFAR-100. The estimation model is Wide-ResNet 40-2 with AutoCleansing. Because $\hat{\alpha}_1$ is fixed to zero, these figures show the standardized values of $\hat{\alpha}_k - \text{Mean}\{\hat{\alpha}_1, \cdots, \hat{\alpha}_K\}$. Let $\hat{\alpha}_n^{Max}$ be the maximum value and $\hat{\alpha}_n^{Min}$ be the minimum value of the standardized sample-category-specific constants for $n$th data. MaxRank is the percentile rank of sorted $\hat{\alpha}_n^{Max}$ in descending order, and MinRank is sorted $\hat{\alpha}_n^{Min}$ in ascending order.

#1 in Figure 4 is an example of incorrect labels within the category set. The original label of #1 is DOG and an alternative to this image is CAT. The estimated $\hat{\alpha}$ of an original label of #1 is 0.233, whereas $\hat{\alpha}$ of an alternative label of #1 is -0.227. Similarly, the original labels are positive $\hat{\alpha}$, whereas alternative labels are negative $\hat{\alpha}$ for images #2-#5 in this figure.

#6 in Figure 5 shows an example of incorrect labels outside the category set. The original label of #6 is TRUCK, whereas the correct label might be PERSON, which does not belong to the category set for the CIFAR-10 dataset. For this image, $\hat{\alpha}$ of the original labels have positive. For these examples of #6-#10, the estimated $\hat{\alpha}$ of the original labels are positive.

#11 in Figure 6 presents an example of multiple objects. The original label of #11 is DEER; however, this image includes an additional object of PERSON that does not belong to the category set of the CIFAR-10 dataset. Images of #12-#15 are examples of multiple objects in the CIFAR-100 datasets. For example, the original label of #12 is PLANE; however, this image also has the object of SEA. Both PLANE and SEA belong to the category set. For these images, $\hat{\alpha}$ of the original labels have positive, $\hat{\alpha}$ and additional labels have negative labels.

Note that the values of MaxRank or MinRank are less than 0.1 % for all images in these figures. This suggests that the mislabeled samples can be identified using the high or low value of the sample-category-specific constants $\hat{\alpha}$ estimated by AutoCleansing. To identify the mislabeled samples, we must specify the threshold criteria between the correct and incorrect labels. Let $\tau$ be the percentage of mislabeled samples in the dataset. Algorithm 2 in Appendix A.4 provides the procedure for searching for mislabeled samples given $\tau$ using AutoCleansing.

After searching for the mislabeled samples, we can remove the mislabeled data from the sample and run the learning models using the trimmed sample. Figure 3 shows the test accuracies of the base model using trimmed data with different threshold criteria of incorrect labels on the CIFAR-10

and CIFAR-100 datasets. The base model is a Wide-ResNet 40-2 model with AutoAugmentation. As can be observed, the test accuracies are highest when 0.2 % of incorrect labels are dropped in both datasets. This figure shows that dropping mislabeled samples at an appropriate drop rate can improve the classification accuracy. However, it is necessary to repeat learning several times with different criteria to determine the optimum drop rate of mislabeled samples. If an excessively small sample is dropped, the effects of bias due to incorrect labels might remain. If excess samples are removed, the estimation efficiency could be reduced because of the decreasing sample size.

Notably, the maximum test accuracies using trimmed data are very close to the accuracies of Auto-Cleansing in this figure. This suggests that AutoCleansing can remove the biased effects of incorrect labels without dropping the mislabeled samples from the datasets. Furthermore, AutoCleansing does not need to repeat learning because it does not require the threshold criteria of the drop rate for the mislabeled samples. Instead of dropping the mislabeled samples that requires the threshold criteria of incorrect labels, AutoCleansing drops the sample-category specific constantsby capturing the mislabeled bias.

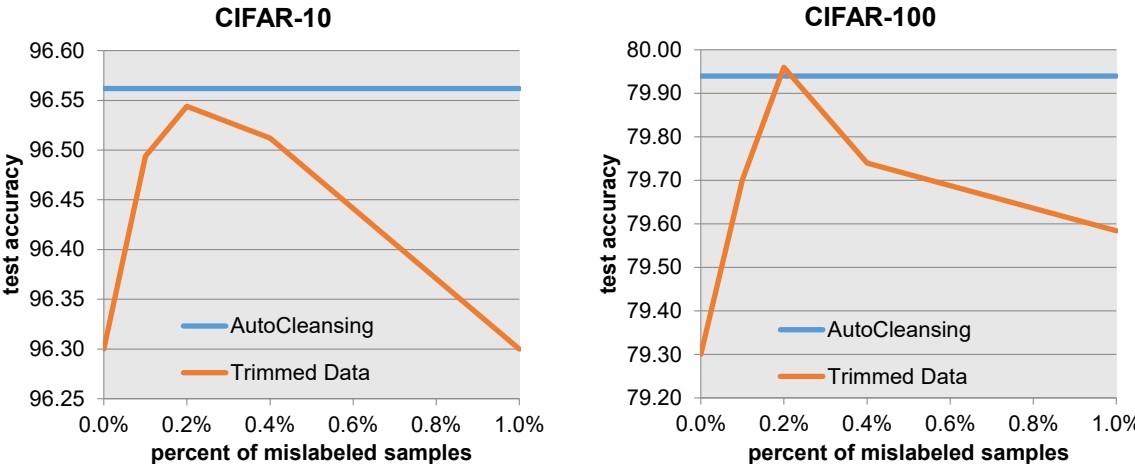

Figure 3: Test accuracies of the base model with trimmed data and AutoCleansing. Averages of five runs are reported.

## 5 CONCLUSION

This study introduces AutoCleansing to address the biased problem due to incorrect labels. The proposed method is appealing in that it can automatically capture the effect of incorrect labels and mitigate it without removing mislabeled samples. As shown in the theoretical analysis, if the model is correctly constructed, the gradient of the expected loss function of AutoCleansing is equal to zero at true parameter values using mislabeled samples for incorrect labels within or outside the category set as well as multiple objects. Furthermore, AutoCleansing can be implemented with any network model and any augmentation method. Experimental results show that the proposed AutoCleansing has better performance than previous studies on CIFAR-10, CIFAR-100, SVHN, and ImageNet datasets. Additional topics for future investigation into AutoCleansing include applications to artificial label noise (Algan & Ulusoy, 2019), the use of other network models such as EfficientNet (Tan & Le, 2019), and the use of recent augmentation methods such as adversarial AutoAugment (Zhang et al., 2019a).

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

# A APPENDIX

## A.1 PROOF OF THEOREM 1

Consider following loss function:

$$\mathcal{L}(\theta) = -\frac{1}{N} \sum_{n=1}^{N} \sum_{i \in \mathbb{K}} 1\left[y_n = i\right] \ln P\left(y_n = i | x_n, \theta\right)$$

where $1[y_n = i]$ is an indicator function equal to one when $y_n = i$ and zero otherwise. Let $L(\theta)$ be the probability limit of the loss function $\mathcal{L}(\theta)$. Consider that the sample has incorrect labels and the outside of the category set is defined as definition 1. If the model is correctly constructed, by the strong law of large numbers, as $N \to \infty$, $L(\theta)$ is as follows:

$$\mathcal{L}(\theta) \xrightarrow{a.s.} L(\theta) = -\mathbb{E}_{x \sim p(X)} \left( \sum_{i \in \mathbb{K}} \left[ Q\left(i \mid x, \mathbb{K}^*, \theta^*\right) \cdot \ln P\left(i \mid x, \mathbb{K}, \theta\right) \right] \right)$$

$$= -\mathbb{E}_{x \sim p(X)} \sum_{\hat{i} \in \mathbb{K}} \left[ \sum_{k \in \mathbb{K}^*} \left[ \pi(i|k, x) \cdot P\left(k \mid x, \mathbb{K}^*, \theta^*\right) \right] \cdot \ln P\left(i \mid x, \mathbb{K}, \theta\right) \right].$$

The derivative of $L(\theta)$ is as follows:

$$\frac{\partial L(\theta)}{\partial \theta} = -\mathbb{E}_{x \sim p(X)} \sum_{i \in \mathbb{K}} \left( \sum_{k \in \mathbb{K}^*} \left[ \frac{\pi(i|k, x) \cdot P\left(k \mid x, \mathbb{K}^*, \theta^*\right)}{P\left(i \mid x, \mathbb{K}, \theta\right)} \right] \frac{\partial P\left(i \mid x, \mathbb{K}, \theta\right)}{\partial \theta} \right)$$

If the distribution function is the softmax function (1), the derivative of $L(\theta)$ can be expressed as follows:

$$\frac{\partial L(\theta)}{\partial \theta} = -\mathbb{E}_{x \sim p(X)} \sum_{i \in \mathbb{K}} \left( \frac{\sum_{j \in \mathbb{K}} e^{m_j}}{e^{m_i} \sum_{j \in \mathbb{K}^*} e^{m_j^*}} \sum_{k \in \mathbb{K}^*} \left[ \pi(i|k, x) \cdot e^{m_k^*} \right] \frac{\partial P\left(i \mid x, \mathbb{K}, \theta\right)}{\partial \theta} \right)$$

where $m_k^* = m_k(x, \theta^*)$ denotes the $k$th element of the output for the base model given the true parameter and the true category set. First, consider the case of no incorrect label, namely, $\pi(i|i, x) = 1$ for all $i \in \mathbb{K}$ and $\mathbb{K} = \mathbb{K}^*$. For this case, the derivative of $L(\theta)$ is equal to zero at the true parameter $\theta = \theta^*$, as follows:

$$\frac{\partial L(\theta^*)}{\partial \theta} = -\mathbb{E}_{x \sim p(X)} \sum_{i \in \mathbb{K}} \left( \frac{\sum_{j \in \mathbb{K}} e^{m_j^*}}{\sum_{j \in \mathbb{K}^*} e^{m_j^*}} \frac{\partial P\left(i \mid x, \mathbb{K}, \theta^*\right)}{\partial \theta} \right)$$

$$= -\mathbb{E}_{x \sim p(X)} \left( \frac{\sum_{j \in \mathbb{K}} e^{m_j^*}}{\sum_{j \in \mathbb{K}^*} e^{m_j^*}} \frac{\partial \sum_{i \in \mathbb{K}} P\left(i \mid x, \mathbb{K}, \theta^*\right)}{\partial \theta} \right) = 0$$

Note that $\sum_{i \in \mathbb{K}} P\left(i \mid x, \mathbb{K}, \theta^*\right) = 1$. From the assumption of identification for the parameter, $L(\theta) \neq L(\theta^*)$ for all $\theta \neq \theta^*$. Thus, if there is no incorrect label and the model is constructed correctly, the estimated parameter $\hat{\theta}$ using the minimum loss function is consistent as follows: $\hat{\theta} \to \theta^*$ as $N \to \infty$. This is a well-known property of consistency in the maximum likelihood estimator of the logit model (Amemiya, 1985).

However, in general, the derivative of $L(\theta)$ is not always equal to zero at the true parameter value if the sample has an incorrect label. If there are mislabeled samples, the estimated parameter $\hat{\theta}$ using the minimum loss function may not converge to the true value.

Consider the estimation using the AutoCleansing model, $c_j(x, \Theta) = m_j(x, \theta) + \alpha_j$ where $\Theta = \{\theta, \alpha\}$. The probability limit of the loss function for the AutoCleansing model is as follows:

$$L^C(\Theta) = -\mathbb{E}_{x \sim p(X)} \sum_{i \in \mathbb{K}} \left[ \sum_{k \in \mathbb{K}^*} \left[ \pi(i|k, x) \cdot P\left(k \mid x, \mathbb{K}^*, \theta^*\right) \right] \cdot \ln P^C\left(i \mid x, \mathbb{K}, \Theta\right) \right].$$

Assume that the probability distribution is a softmax function (2). Let $\theta^+$ be the set of the solution to $\partial L^C / \partial \theta = 0$. The derivative of $L^C(\Theta)$ can be expressed as follows:

$$\frac{\partial L^C(\Theta)}{\partial \theta} = -\mathbb{E}_{x \sim p(X)} \sum_{i \in \mathbb{K}} \left( \frac{\sum\limits_{j \in \mathbb{K}} e^{m_j + \alpha_j}}{\sum\limits_{j \in \mathbb{K}^*} e^{m_j^*}} \frac{\sum\limits_{k \in \mathbb{K}^*} \left[ \pi(i|k,x) \cdot e^{m_k^*} \right]}{e^{m_i + \alpha_j}} \frac{\partial P^C(i \mid x, \mathbb{K}, \Theta)}{\partial \theta} \right)$$

$$= -\mathbb{E}_{x \sim p(X)} \sum_{i \in \mathbb{K}} \left( \frac{\sum\limits_{j \in \mathbb{K}} e^{m_j + \alpha_j}}{\sum\limits_{j \in \mathbb{K}^*} e^{m_j^*}} \frac{e^{m_i^* + \alpha_i^*}}{e^{m_i + \alpha_j}} \frac{\partial P^C(i \mid x, \mathbb{K}, \Theta)}{\partial \theta} \right)$$

where $\alpha_i^* = \ln \left( \sum\limits_{k \in \mathbb{K}^*} \pi(i|k,x) \cdot e^{m_k^*} \right) - m_i^*$. Thus, the derivative of $L^C(\Theta)$ is equal to zero at $\Theta^* = \{\theta^*, \alpha^*\}$. Namely, $\theta^* \in \theta^+$. $\qquad \square$

## A.2 PROOF OF THEOREM 2

From the assumption, $Q(f \mid x, \mathbb{K}^*, \theta^*) \geq Q(t \mid x, \mathbb{K}^*, \theta^*)$. If the model is correctly constructed, the output of the model for the true category is higher than that of the other category. Therefore, $m_t^* \geq m_j^*, \forall j$. For the general case, from Theorem 1, the difference between the sample-category-specific constants of the false label and that of the true label is as follows:

$$\alpha_f^* - \alpha_t^* = \ln \left( \sum_{k \in \mathbb{K}^*} \pi(f|k,x) \cdot e^{m_k^*} \right) - m_f^* - \left[ \ln \left( \sum_{k \in \mathbb{K}^*} \pi(t|k,x) \cdot e^{m_k^*} \right) - m_t^* \right]$$

$$= \ln \frac{\sum\limits_{k \in \mathbb{K}^*} \pi(f|k,x) \cdot \frac{e^{m_k^*}}{\sum\limits_{j \in \mathbb{K}^*} e^{m_j^*}}}{\sum\limits_{k \in \mathbb{K}^*} \pi(t|k,x) \cdot \frac{e^{m_k^*}}{\sum\limits_{j \in \mathbb{K}^*} e^{m_j^*}}} - m_f^* + m_t^*$$

$$= \ln \frac{Q(f \mid x, \mathbb{K}^*, \theta^*)}{Q(t \mid x, \mathbb{K}^*, \theta^*)} - m_f^* + m_t^* \geq 0.$$

For the symmetric case, $\pi(t|k,x) = \pi(k|t,x)$ for all $k$ Therefore, the sample-category-specific constants of the true label are as follows:

$$\alpha_t^* = \ln \left( \sum_{k \in \mathbb{K}^*} \pi(t|k,x) \cdot e^{m_k^*} \right) - m_t^* = \ln \left( \sum_{k \in \mathbb{K}^*} \pi(k|t,x) \cdot e^{m_k^*} \right) - m_t^*$$

$$\leq \ln \left( \sum_{k \in \mathbb{K}^*} \pi(k|t,x) \cdot e^{m_t^*} \right) - m_t^* = 0.$$

For the single-symmetric case, $\pi(j|j) = 1 \, \forall j \neq f, t$. Therefore, the sample-category-specific constants of category $j \neq f, t$ are as follows:

$$\alpha_j^* = \ln \left( \sum_{k \in \mathbb{K}^*} \pi(j|k,x) \cdot e^{m_k^*} \right) - m_j^* = \ln \left( \sum_{k \in \mathbb{K}^*} \pi(k|j,x) \cdot e^{m_k^*} \right) - m_j^* = 0$$

$$\square$$

## A.3 ADDITIONAL TABLES

Table 7: Hyperparameters for the Experiment. LR represents the learning rate, whereas WD represents the weight decay. Multi steps schedule decays the learning rate by 10-fold at epochs (150, 225) for CIFAR and (90, 180, 240) for ImageNet.

| Dataset | Model | LR | LR Schedule | WD | Batch Size | Epoch |
|---------|-------|-----|-------------|-----|------------|-------|
| CIFAR-10 | ResNet-32 | 0.1 | multi steps | 0.0001 | 256 | 300 |
| CIFAR-10 | Wide-ResNet-40-2 | 0.1 | cosine | 0.0002 | 128 | 200 |
| CIFAR-10 | Wide-ResNet-28-10 | 0.1 | cosine | 0.0005 | 128 | 200 |
| CIFAR-10 | Shake-Shake (26 2x32d) | 0.01 | cosine | 0.001 | 128 | 1800 |
| CIFAR-10 | Shake-Shake (26 2x96d) | 0.01 | cosine | 0.001 | 128 | 1800 |
| CIFAR-10 | Shake-Shake (26 2x112d) | 0.01 | cosine | 0.001 | 128 | 1800 |
| CIFAR-10 | PyramidNet+ShakeDrop | 0.05 | cosine | 5.00E-05 | 64 | 1800 |
| CIFAR-100 | ResNet-32 | 0.1 | multi steps | 0.0001 | 256 | 300 |
| CIFAR-100 | Wide-ResNet-28-10 | 0.1 | cosine | 0.0005 | 128 | 200 |
| CIFAR-100 | Shake-Shake (26 2x96d) | 0.01 | cosine | 0.0025 | 128 | 1800 |
| CIFAR-100 | PyramidNet+ShakeDrop | 0.025 | cosine | 0.0005 | 64 | 1800 |
| SVHN | Wide-ResNet-28-10 | 0.005 | cosine | 0.001 | 128 | 200 |
| ImageNet | Resnet-50 | 0.1 | multi steps | 0.0001 | 256 | 270 |

Table 8: AutoCleansing with the sample-category-specific constant (AC1) and the sample specific constant (AC2). Averages of five runs are reported.

| | | AA | AC1+AA | | AC2+AA | |
|---|---|-----|--------|---|--------|---|
| CIFAR-10 | Wide-ResNet-40-2 | 96.30 | **96.56** | ±0.11 | 96.54 | ±0.11 |
| | Wide-ResNet-28-10 | 97.32 | **97.53** | ±0.08 | 97.48 | ±0.06 |
| CIFAR-100 | Wide-ResNet-40-2 | 79.30 | **79.94** | ±0.20 | 79.86 | ±0.19 |
| | Wide-ResNet-28-10 | 82.91 | 84.07 | ±0.18 | **84.12** | ±0.17 |

## A.4 ALGORITHMS

---

**Algorithm 1** AutoCleansing

**Require:** Initial parameter $\alpha \in \mathbb{R}^{N \times K}$ of the sample-category specific constant. $N$: The number of samples. $K$: The number of categories. $B$: The number of minibatches.
**Learning:**
**for** $b = 1, \ldots, B$ **do**

    Sample a minibatch of $x^{(b)}$ from training data set $X$. Let $d^{(b)}$ be a vector of index of the sampling such that $x^{(b)} = X[d^{(b)}]$.

    Compute gradient using the cleansing model for the minibatch $C(x^{(b)}, \theta, \alpha) = M(x^{(b)}, \theta) + \alpha^{(b)}$, where $\alpha^{(b)} = \alpha[d^{(b)}] \in \mathbb{R}^{N_b \times K}$ is the constant for the minibatch and $N_b$ is the batch size.

    Update the parameter $\Theta = \{\theta, \alpha\}$.

**end for**
**Testing:** Predict the category using the cleansed model $M(x, \hat{\theta}) = C(x, \hat{\theta}, \hat{\alpha}) - \hat{\alpha}$.

---

**Algorithm 2** Detection of mislabeled samples using AutoCleansing

**Require:** $\hat{\alpha} \in \mathbb{R}^{N \times K}$: the sample-category specific constants of $n$th sample and $k$th category, $N$: the number of samples, $K$: the number of categories, and $\tau$: the percent of mislabeled samples.
**Require:** Standardize $\hat{\alpha}_{nk} \leftarrow \hat{\alpha}_{nk} - \text{Mean}\{\hat{\alpha}_{n1}, \cdots, \hat{\alpha}_{nK}\}$ for all $n = \{1, \cdots, N\}$.
**Require:** Set $\hat{\alpha}_n^{Max} = \text{Max}\{\hat{\alpha}_{n1}, \cdots, \hat{\alpha}_{nK}\}$ and $\hat{\alpha}_n^{Min} = \text{Min}\{\hat{\alpha}_{n1}, \cdots, \hat{\alpha}_{nK}\}$ for all $n = \{1, \cdots, N\}$.
**Require:** Set $L_{miss} = []$: mislabeled list. $N_{miss} = \tau N/100$: the number of mislabeled samples.
**while** the size of $L_{miss} < N_{miss}$ **do**

    Find the sample index $n$ such as $\hat{\alpha}_n^{Max} = \text{Max}(\hat{\alpha}^{Max})$. Delete $\hat{\alpha}_n^{Max}$ from $\hat{\alpha}^{Max}$. Add the sample index $n$ to the mislabeled list $L_{miss}$ if $n \notin L_{miss}$.

    **if** the size of $L_{miss} < N_{miss}$ **do**

        Find the sample index $n$ such as $\hat{\alpha}_n^{Min} = \text{Min}(\hat{\alpha}^{Min})$. Delete $\hat{\alpha}_n^{Min}$ from $\hat{\alpha}^{Min}$. Add the sample index $n$ to the mislabeled list $L_{miss}$ if $n \notin L_{miss}$

    **end if**

**end while**
**Return:** the mislabeled list $L_{miss}$.

---

## A.5 ADDITIONAL FIGURES

| Dataset ID | Image | Origianl labels $\alpha_f$ | Alternative labels $\alpha_t$ | MaxRank | MinRank |
|---|---|---|---|---|---|
| CIFAR-10 #1 38775 | | dog 0.233 | cat -0.227 | 2.002% | 0.028% |
| CIFAR-10 #2 25684 | | cat 0.274 | dog -0.235 | 0.438% | 0.020% |
| CIFAR-100 #3 10178 | | leopard 0.337 | tiger -0.300 | 0.020% | 0.002% |
| CIFAR-100 #4 7019 | | tulip 0.312 | rose -0.240 | 1.328% | 0.038% |
| CIFAR-100 #5 24900 | | cloud 0.334 | can -0.182 | 0.072% | 0.268% |

Figure 4: Example images of incorrect labels within category set and sample-category specific constants $\alpha$. MaxRank is the percentile rank of sorted $\hat{\alpha}_n^{Max}$ in descending order, and MinRank is sorted $\hat{\alpha}_n^{Min}$ in ascending order. See text for more details.

| | Dataset

ID | Image | Origianl labels
$\alpha_f$ | Alternative labels
$\alpha_t$ | MaxRank | MinRank |
|---|---|---|---|---|---|---|
| #6 | CIFAR-10

18310 | | truck

0.332 | person

- | 0.002% | 1.242% |
| #7 | CIFAR-100

19011 | | lion

0.335 | giraffe

- | 0.044% | 39.838% |
| #8 | CIFAR-100

1578 | | plate

0.335 | cutlery

- | 0.054% | 13.646% |
| #9 | CIFAR-100

29783 | | cup

0.338 | glass

- | 0.016% | 43.730% |
| #10 | CIFAR-100

48760 | | motorcycle

0.345 | raincoat

- | 0.002% | 7.970% |

Figure 5: Example images of incorrect labels outside category set and sample-category specific constants $\alpha$.

| | Dataset ID | Image | Origianl labels $\alpha_f$ | Alternative labels $\alpha_t$ | MaxRank | MinRank |
|---|---|---|---|---|---|---|
| #11 | CIFAR-10

3879 |  | deer

0.317 | person

- | 0.024% | 0.734% |
| #12 | CIFAR-100

16305 |  | plain

0.332 | sea

-0.281 | 0.112% | 0.006% |
| #13 | CIFAR-10

9145 |  | cat

0.247 | dog

-0.212 | 1.356% | 0.060% |
| #14 | CIFAR-100

33026 |  | bus

0.323 | road

-0.231 | 0.450% | 0.052% |
| #15 | CIFAR-100

780 |  | house

0.284 | road

-0.218 | 5.558% | 0.088% |

Figure 6: Example images of incorrect labels with multiple objects and sample-category specific constants $\alpha$.

