# OpenReview forum: "AutoCleansing: Unbiased Estimation of Deep Learning with Mislabeled Data"
_ICLR.cc/2021/Conference — Reject_

### Official Review · AnonReviewer2 · 2020-10-25
**Serious Flaw in Problem Formulation and Subsequent Theorems**

**Rating:** 3
**Confidence:** 5

**Review:**

### Paper Summary

In this paper, the authors proposed to train high quality classifiers from datasets with some mislabels.

For this purpose, the authors considered adjusting the softmax prediction using an additional term $\alpha$ as follows:
$$
P^C(i | x; \theta) = \frac{\exp(m_i(x; \theta) + \alpha_i(x))}{\sum_j \exp(m_j(x; \theta) + \alpha_j(x))}
$$
where $m(\cdot; \theta)$ is a model and $\theta$ is its parameter.
The authors claimed that, by adjusting $\alpha$ through training, the trained model $m(\cdot; \hat{\theta})$ with an optimal parameter $\hat{\theta}$ is asymptotically consistent with the model trained on a dataset with clean labels, i.e.,  the trained model without $\alpha$ performs well on clean test data
$$
P(i | x; \theta) =\frac{\exp(m_i(x; \theta))}{\sum_j \exp(m_j(x; \theta))}
$$
In the proposed method, for the training set $D = \\{x_n, y_n \\}$, we first train the model by minimizing the following loss function:
$$
\hat{\theta}, \hat{\alpha} = \arg\min_{\theta \in \Theta, \alpha \in \mathbb{R}^{N \times K}} -\frac{1}{N} \sum_{n=1}^N \sum_{i=1}^K 1[y_n = i] \log \frac{\exp(m_i(x_n; \theta) + \alpha_{ni})}{\sum_j \exp(m_j(x_n; \theta) + \alpha_{nj})}
$$
where $K$ is the number of classes. We then classify the new instance $x$ by $\hat{y} =\arg\max_i m_i(x; \hat{\theta})$.

In Theorem 1, the authors claimed that the above estimator $\hat{\theta}$ converges to the *true* parameter $\theta^*$.



### Pros & Cons

[Pros]

The experimental results indicate that the proposed method is effective on several datasets.

[Cons]

The paper contains a serious flaw in its problem formulation and the subsequent theorems. The proposed problem formulation has a trivial solution which is completely useless. The effectiveness reported in the experiments seems to be just an artifact caused by the tunings of hyperparameters. See my comments in "Quality" below for the detail.



### Quality

The paper contains a serious flaw in its problem formulation.

Recall the training problem:
$$
\hat{\theta}, \hat{\alpha} = \arg\min_{\theta \in \Theta, \alpha \in \mathbb{R}^{N \times K}} -\frac{1}{N} \sum_{n=1}^N \sum_{i=1}^K 1[y_n = i] \log \frac{\exp(m_i(x_n; \theta) + \alpha_{ni})}{\sum_j \exp(m_j(x_n; \theta) + \alpha_{nj})}
$$
This problem has a trivial solution that $\alpha_{n y_n} \to +\infty$ for $\forall n$, which leads to
$$
\frac{\exp(m_i(x_n; \theta) + \alpha_{ni})}{\sum_j \exp(m_j(x_n; \theta) + \alpha_{nj})} \to
\delta(y_n = i)
$$
where $\delta(y_n = i) = 1$ if $y_n=i$ and 0 otherwise.
Note that this trivial solution does not depend on the model $m(\cdot; \theta)$. Thus, any parameter $\theta$ can be an optimal solution $\hat{\theta}$ as long as $m(\cdot ;\theta)$ is finite.

The above observation indicates that the proposed method do not work as expected if the training problem is solved appropriately. Thus, I conjecture that the good performances reported in the experiments are the artifact caused by the tuning of hyperparameters, e.g., the training converged to local optima that occasionally performed well.

Note that the above observation on the training problem also suggests that the claim of Theorem 1 (the estimator $\hat{\theta}$ converges to the *true* parameter $\theta^*$) is not correct.

In the proof, the authors considered the following objective function:
$$
L^C(\theta, \alpha) = - \mathbb{E} \sum_{i=1}^K \left[ \sum_{k=1}^K \pi(i | k, x) P(k | x, \theta^*) \log P^C(i | x, \theta) \right]
$$
Let $U(i | x, \theta^*) = \sum_{k=1}^K \pi(i | k, x) P(k | x, \theta^*)$. We then have
$$
L^C(\theta, \alpha) = - \mathbb{E} \sum_{i=1}^K U(i | x, \theta^*) \log \frac{\exp(m_i(x; \theta) + \alpha_i(x))}{\sum_j \exp(m_j(x; \theta) + \alpha_j(x))}
$$
By taking the derivative with respect to $\omega \in \\{\theta, \alpha\\}$, we have
$$
\frac{\partial L^C(\theta, \alpha)}{\partial \omega} = - \mathbb{E} \sum_{i=1}^K U(i | x, \theta^*)\left( \frac{\partial (m_i(x; \theta) + \alpha_i(x))}{\partial \omega} - \sum_{k=1}^K \frac{\exp(m_k(x; \theta) + \alpha_k(x))}{\sum_j \exp(m_j(x; \theta) + \alpha_j(x))} \frac{\partial (m_k(x; \theta) + \alpha_k(x))}{\partial \omega} \right) \\
= - \mathbb{E} \sum_{i=1}^K \left( U(i | x, \theta^*) - \frac{\exp(m_i(x; \theta) + \alpha_i(x))}{\sum_j \exp(m_j(x; \theta) + \alpha_j(x))} \right) \frac{\partial (m_i(x; \theta) + \alpha_i(x))}{\partial \omega}
$$
Thus, any $\theta, \alpha$ that satisfy $U(i | x, \theta^*) = \frac{\exp(m_i(x; \theta) + \alpha_i(x))}{\sum_j \exp(m_j(x; \theta) + \alpha_j(x))}$ are optimal.

In the proof of Theorem 1, the authors only considered a specific $\alpha$, and overlooked the existence of other $\alpha$ that are equally optimal, which led to the wrong claim that $\hat{\theta}$ converges to $\theta^*$.



### Clarity

Apart from the serious flaw above, I think the paper is clearly written and the main claim of the paper is easy to follow.



### Originality

The use of the adjustable parameters for fitting noisy data is studied in the literature of robust learning. I would like to suggest the authors to see [Ref1] and references therein. In [Ref1], an additional penalty is imposed on the adjustable parameter to avoid the trivial solution I raised above.

[Ref1] Consistent Robust Regression, NeurIPS17.



### Significance

Because of the flaw I raised above, I think the contribution of this paper is not significant.


----------
### Feedback after discussion

I would like to clarify my thought here. Recall that using the weight decay is equivalent to adding the L2 regularization to the training objective. An important observation here is that the addition of the L2 regularization to the proposed objective will make the global optima non-trivial (apart from the trivial ones I raised), and there might be a hope that the new global optima has some useful properties. What this observation indicates is that the use of the L2 regularization (or weight decay) is an essential factor for the proposed method to output something meaningful. This fact also implies that the analysis of the objective function alone (without the regularization, in Section3) is no longer meaningful. Moreover, because the L2 regularization (or weight decay) is an essential factor, the tuning of its weight should have a major impact to the resulting model. I therefore think it will be important to investigate the effect of such a weight in the experiments, instead of just using a standard weight.

---

> ### Author Response · Authors · 2020-11-17
> **Thank you for the helpful comment. The description of the theorem is revised.**
>
> > Note that the above observation on the training problem also suggests that the claim of Theorem 1 (the estimator $\theta$ converges to the true parameter $\theta^∗$ is not correct.
>
> As you pointed out, while this theorem states that the loss function with AutoCleansing is minimized locally at true values, it does not guarantee that minimization of the loss function will converge to the true value, if the loss function has more than one local minimum. Following your suggestion, I have removed the description of the convergence to true value from Theorem 1 and added the note that the loss function with AutoCleansing is minimized locally at true values.
>
> > The above observation indicates that the proposed method do not work as expected if the training problem is solved appropriately. Thus, I conjecture that the good performances reported in the experiments are the artifact caused by the tuning of hyperparameters, e.g., the training converged to local optima that occasionally performed well.
>
> As mentioned in page 6, all hyperparameters are same as those used in previous works. I did not tune the hyperparameters.
>
> > The use of the adjustable parameters for fitting noisy data is studied in the literature of robust learning. I would like to suggest the authors to see [Ref1] and references therein. In [Ref1], an additional penalty is imposed on the adjustable parameter to avoid the trivial solution I raised above.
>
> To avoid the trivial solution, I have implemented the weight decay for $\alpha$ that imposes additional penalty. I have added the description on the regularization in page 6. Following your suggestion, I have referred to the consistent robust regression in section 2.
>
> I believe that I have addressed your concerns. Again, thank you for giving me the opportunity to strengthen the manuscript with your valuable comments.

---

> > ### Comment · AnonReviewer2 · 2020-11-22
> > **What is the justification of preferring a local optima rather than the global optima?**
> >
> > ### Major concern is not resolved.
> >
> > As I pointed out in my review, the global optima of the proposed training objective is useless.
> > The question is why we need to train models using such a flawed objective.
> > I understand that the authors used weight decay to avoid global optima, which might have led to a better local optima.
> > That is, we should not globally optimize the objective, and we instead hope an optimizer to find a good local optima.
> > What is the role of the objective function if we are not allowed to optimize it globally?
> > What is the justification of preferring a local optima rather than the global optima?
> >
> >
> > ### The revision seems to be not appropriate.
> >
> > I briefly checked the revision, and found Theorem 1 is still not appropriate.
> > There is no definition what "minimized locally" stands for.
> >
> > Moreover, the proof still says that $\Theta = \Theta^*$ which is not correct.
> > As I pointed out,  any $\theta, \alpha$ that satisfy $U(i | x, \theta^*) = \frac{\exp(m_i(x; \theta) + \alpha_i(x))}{\sum_j \exp(m_j(x; \theta) + \alpha_j(x))}$ are optimal.
> > That is, there is no unique global optima.
> > All we have is the set of global optimal solutions $\Theta^* = \\left\\{ \theta, \alpha \mid U(i | x, \theta^*) = \frac{\exp(m_i(x; \theta) + \alpha_i(x))}{\sum_j \exp(m_j(x; \theta) + \alpha_j(x))} \\right\\}$.
> > Hence, the only thing what we might be able to claim would be $\theta, \alpha \in \Theta^*$ as $N \to \infty$.

---

> > > ### Author Response · Authors · 2020-11-24
> > > **Local minima are no longer considered to be a serious problem for neural network optimization**
> > >
> > > Thank you for additional comments for my revised paper.
> > >
> > > > What is the justification of preferring a local optima rather than the global optima?
> > >
> > > Recent theoretical and empirical results strongly suggest that local minima are no longer considered to be a serious problem for neural network optimization (LeCun et al., 2015; Goodfellow et al., 2016, p529). In this study, I have confirmed the very similar results for five runs with different initial values using the CIFAR-10/100 datasets. Furthermore, the weight decay is a standard procedure to avoid over-fitting in the neural network models.
> > >
> > > Yann LeCun , Yoshua Bengio, and Geoffrey Hinton (2015) Deep Learning. Nature
> > >
> > > Ian Goodfellow, Yoshua Bengio, and Aaron Courville Goodfellow (2016) Deep Learning. MIT Press.
> > >
> > > > I briefly checked the revision, and found Theorem 1 is still not appropriate. There is no definition what "minimized locally" stands for.
> > >
> > > Following your suggestion, I have removed the description “minimized locally” and revised the Theorem 1, as the gradient of the expected loss function with AutoCleansing is zero at the true parameter value.
> > >
> > > > Hence, the only thing what we might be able to claim would be $\theta, \alpha \in \Theta^*$ as $N \mathrm{\to \infty }$
> > >
> > > Following your comments, I have added the description of $\theta^* \in \theta^+$ in Theorem 1, where $\theta^+$ is the set of the solution to ${\partial L^C}/ {\partial \theta} = 0$.
> > >
> > >
> > > Your comments are very helpfull to clarify the description of the theoretical section of my paper.
> > > Thanks again.

---

### Official Review · AnonReviewer1 · 2020-10-28
**Interesting and simple idea but lacks comparison with any other methods for learning with noisy labels**

**Rating:** 4
**Confidence:** 4

**Review:**

The paper proposes an approach for handling noisy labels in predictive models without removing them. The approach is based on a base network and a category dependent constant. At test time the prediction is done using the base network. The paper is well-written and the ideas are explained clearly. I have the following comments and questions on the empirical and theoretical aspects of the work:

As far as I understand, theoretical analysis is only applicable to AC1 model but the more practical approach which is applicable to datasets such as imagenet is AC2. What portion of your results can be extended to AC2?

The experiments on CIFAR-10, CIFAR-100 and SVHN do not show a statistically significant improvement over the baselines; specifically after considering the confidence interval for the AA results (from table 2 of the AutoAugment paper). Adding experiments with synthetic datasets with different levels of noise can be helpful in understanding the advantages of AC1/AC2 over other methods for handling noisy labels.

Experiments in Section 4.2 are only trimming the incorrect training labels; ideally, you also want to remove the noisy labels from the test set too. However, as far as I understand it’s not straightforward to apply AC1 to the test set and trim the noisy labels. Is this correct?

Other methods for learning with noisy labels such as the ones that are mentioned in the Section 2: ‘Related Works’ should be added in order to provide a better picture of the pros and cons of the method. In the current version of the paper, the emphasis in the experiments is on augmentation methods which may not be the best choice.  For instance, “Unsupervised Label Noise Modeling and Loss Correction” by Arazo et al. has a similar approach to AC1 as it also models each sample and doesn’t require a matrix based noise model. This could be a good candidate baseline.

Overall, I like the simplicity of the approach but I believe with the current state of the paper, it’s hard to judge the value of the approach over other methods for learning with noisy labels.

---

> ### Author Response · Authors · 2020-11-17
> **The experiment on comparison with other method has been added.**
>
> > As far as I understand, theoretical analysis is only applicable to AC1 model but the more practical approach which is applicable to datasets such as imagenet is AC2. What portion of your results can be extended to AC2?
>
> AC2 corresponds to the special case of the single symmetric case in Theorem 2. For the single symmetric case, $ \alpha_{nj}=0 ~\forall n,\forall j\ne t,f $. If the true category belongs to the outside of the category set, we cannot estimate the $ \alpha_{nt} $ of the true category, therefore, all categories except the observed label have zero values of $ \alpha_{nj} $ . I have added the description in page 6.
>
> > The experiments on CIFAR-10, CIFAR-100 and SVHN do not show a statistically significant improvement over the baselines; specifically after considering the confidence interval for the AA results (from table 2 of the AutoAugment paper).
>
> My experimental results of the proposed AC1+AA show a significant improvement over AA results in most network models. The confidence interval of AutoAugment paper might be rounded up. For example, test error of Wide-Res-Net 28-10 for CIFAR 10 dataset is 2.6±0.1 in AutoAugment, while 2.58 ± 0.062 in PBA.
>
> > Adding experiments with synthetic datasets with different levels of noise can be helpful in understanding the advantages of AC1/AC2 over other methods for handling noisy labels.
>
> I agree that the experiments with synthetic datasets can be helpful. However, the purpose of this paper is to analyze incorrect label in the real-world datasets, not artificial label noise. Therefore, I have referred the experiments with synthetic datasets for the future works.
>
> > Experiments in Section 4.2 are only trimming the incorrect training labels; ideally, you also want to remove the noisy labels from the test set too. However, as far as I understand it’s not straightforward to apply AC1 to the test set and trim the noisy labels. Is this correct?
>
> Yes. It might be future works to analyze the noisy labels in test dataset using AC1.
>
> > Other methods for learning with noisy labels such as the ones that are mentioned in the Section 2: ‘Related Works’ should be added in order to provide a better picture of the pros and cons of the method.
>
> Following your suggestion, the pros and cons of related works have been added in section 2.
>
> > In the current version of the paper, the emphasis in the experiments is on augmentation methods which may not be the best choice. For instance, “Unsupervised Label Noise Modeling and Loss Correction” by Arazo et al. has a similar approach to AC1 as it also models each sample and doesn’t require a matrix based noise model. This could be a good candidate baseline.
>
> Thank you for helpful suggestion. I have added the description of Arazo et al. in section 2 and deleted the experiment on augmentation methods. I agree that the comparison with the method proposed by Arazo et al might to be important. However, their experiment considered the synthetic label noise and they did not provide the experiment of incorrect labels in the real-world datasets. Alternatively, the comparison with the area under the margin (AUM) proposed by Pleiss et al. (2020) has been added (Table 6). They provided the experiments of the label noise in the real-world datasets as well as the artificially generated label noise. Their results show that the AUM has better performance than the previous studies for dataset with synthetic label noise added. AutoCleansing demonstrates outperforming the AUM on both CIFAR-10 and CIFAR-100.
>
> Again, thank you for giving me the opportunity to strengthen the manuscript with your valuable comments.

---

### Official Review · AnonReviewer4 · 2020-10-28
**This study provides a theoretical model to capture biased effect of incorrect labels automatically and address the prediction errors due to incorrect labels in training data.**

**Rating:** 6
**Confidence:** 3

**Review:**

This paper seems to be a useful contribution to the literature on deep learning with noisy datasets, showing a good improvement over the state of the art.
This work presents a theoretical model formulation to capture the biased effects of incorrect labels automatically
The paper is generally well-written and structured clearly. However, there are few small changes or suggestions to improve are as follows:
•	The authors have not rationalized enough the performance of the AutoCleansing over the other methods and on the data sets in detail for better readability.
•	Summary of the data sets as a table provides better visibility and readability.
•	Detection of incorrect labels using the proposed method has been described in detail but why AutoCleansing does not require the threshold criteria of the drop rate needs further discussion. Please add additional details.
•	In addition to the learning rate, weight decay, Epoch etc. the authors can add the time units for each of the datasets helps the future researchers. Please add.

The paper and the supplementary provided describes the theoretical formulation and other objective functions in full detail and provides enough information for an expert reader to understand and interpret the results.

---

> ### Author Response · Authors · 2020-11-17
> **The experiment on comparison with other method has been added.**
>
> > • The authors have not rationalized enough the performance of the AutoCleansing over the other methods and on the data sets in detail for better readability.
>
> Following your suggestion, the comparison with the area under the margin (AUM) proposed by Pleiss et al. (2020) has been added (Table 6). They provided the experiments of the label noise in the real-world datasets as well as the artificially generated label noise. Their results show that the AUM has better performance than the previous studies for dataset with synthetic label noise added. AutoCleansing demonstrates outperforming the AUM on both CIFAR-10 and CIFAR-100.
>
> > • Detection of incorrect labels using the proposed method has been described in detail but why AutoCleansing does not require the threshold criteria of the drop rate needs further discussion. Please add additional details.
>
> Instead of dropping the mislabeled samples that requires the threshold criteria of incorrect labels, AutoCleansing drops the sample-category specific constants capturing the mislabeled bias. I have added the description in page 9.
>
> > • In addition to the learning rate, weight decay, Epoch etc. the authors can add the time units for each of the datasets helps the future researchers.
>
> The additional learning time of AutoCleansing with CIFAR-10/100 datasets is only 0.5 % of the learning time of the base network models. I have added the description in page 3.
>
> Again, thank you for giving me the opportunity to strengthen the manuscript with your valuable comments.

---

### Official Review · AnonReviewer3 · 2020-11-04
**The paper method can improve the performance to some extent, but the influence is not prominent.**

**Rating:** 5
**Confidence:** 3

**Review:**

This study introduces AutoCleansing to address the biased problem due to incorrect labels. This framework can automatically capture the effect of incorrect labels and mitigate it without removing mislabeled samples. There is some improvement in performance, but not much difference.

Where to improve:

The challenges of solving the problem of incorrect labels must be explained in-depth, but they are not introduced at all.
Section 2 leaves me with an idea of incompleteness, and I would request the authors to make it a critical review and not just a list of methods.
Section 3 for the proposed method AutoCleansing. The method is primarily described through equations. The authors may want to consider adding more descriptive text (and possibly figures) that would make the paper more accessible to readers without and extensive mathematics background.
the section Results should be substantially expanded. The authors should be explicit about the difference between the model proposed here and the models implemented by previous studies and how their model works compared to other methods.

---

> ### Author Response · Authors · 2020-11-17
> **The numerical example describing the theorem and the experiment on comparison with other method has been added.**
>
> Following your suggestion, a critical review has been added in section 2. For the description of theoretical analysis in section 3, Table 1 has been introduced to illustrate the theorems using the numerical examples. For the experiments, the comparison with the area under the margin (AUM) proposed by Pleiss et al. (2020) has been added (Table 6). They provided the experiments of the label noise in the real-world datasets as well as the artificially generated label noise. Their results show that the AUM has better performance than the previous studies for dataset with synthetic label noise added. AutoCleansing has better performance than AUM on both CIFAR-10 and CIFAR-100.
>
> I believe that I have addressed your concerns. Again, thank you for giving me the opportunity to strengthen the manuscript with your valuable comments.

---

### Author Response · Authors · 2020-11-24
**Summary of Revision**

Thank you for numerous helpful and constructive comments of my paper. A summary of major updates of the revision is as follows:

1. The critical reviews of related works have been added in section 2.
2. The description of the local minimum in Theorem 1 is revised as the gradient of the expected loss function with AutoCleansing is zero at the true parameter value.
3. Table 1 has been introduced to illustrate the theorems using the numerical examples.
4. The comparison with AUM (the area under the margin) for learning with noisy labels has been added (Table 6). AutoCleansing demonstrates outperforming the AUM on both CIFAR-10 and CIFAR-100.

---

### Decision · Program_Chairs · 2021-01-07
**Final Decision**

**Decision:**

Reject

**Comment:**

The paper addresses learning with noisy labels, by detecting and correcting samples with noisy labels. Reviewers had concerns about the empirical evaluations, specifically about comparing to additional methods, about hyperparameter tuning, and about the improvements being vey small. There was also a concern that the analysis of the objective does not take into account explicitly the L2 regularization induced by weight decay. Based on these concerns the paper is not ready yet for publication.